# Short communication: A comparison between two glucose measurement methods in beef steers during a glucose tolerance test

Kirsten R. Nickles[1], Alejandro E. Relling[1‡], Alvaro Garcia-Guerra[2‡], Francis L. Fluharty[3‡], Anthony J. Parker[1]*

1 Department of Animal Sciences, The Ohio State University, Wooster, Ohio, United States of America,
2 Department of Animal Sciences, The Ohio State University, Columbus, Ohio, United States of America,
3 Department of Animal and Dairy Science, University of Georgia, Athens, Georgia, United States of America

☯ These authors contributed equally to this work.
‡ These authors also contributed equally to this work.
* tjparker967@gmail.com

**Data Availability Statement:** All relevant data are within the paper and its Supporting Information files.

## Abstract

Glucose tolerance tests (GTT) are commonly performed in beef cattle to evaluate the glucose-insulin signaling pathway. Blood samples are obtained via a catheter and then transferred back to the laboratory for further analysis. A hand-held glucometer used chute-side can make performing GTT's and quantifying blood glucose concentration much easier and faster for research purposes. The purpose of this study was to evaluate the agreement between a hand-held electronic glucometer (Precision Xtra; Abbott Diabetes Care Inc., Mississauga, ON, Canada) for chute-side use in beef cattle compared with a colorimetric assay in the laboratory (Stanbio Glucose LiquiColor; Stanbio Laboratory, Boerne, TX, USA). A GTT was performed on 13 Simmental × Angus steers during the growing phase. Blood samples were obtained via a jugular catheter. Glucometer readings were taken immediately after blood was sampled from the jugular with no preservative, and laboratory measurements were conducted on plasma preserved with sodium fluoride. A paired t-test ($P = 0.40$), Pearson's correlation ($P < 0.001$; $r = 0.95$), Bland-Altman plot, and Lin's concordance correlation coefficient (LCCC = 0.90) were completed to evaluate the performance of the glucometer relative to the results from the laboratory assay. Based on the results, we conclude that the glucometer is an acceptable method for measuring blood glucose concentration in beef cattle under field conditions.

## Introduction

Glucose tolerance tests (GTT) are commonly used in beef cattle research as an indicator of insulin resistance [1]. This procedure has been modified by several research groups, but generally consists of an intravenous bolus infusion of 50% glucose followed by continuous blood sampling to monitor glucose and insulin concentration. The standard method of measuring glucose in plasma is with a colorimetric assay to quantify glucose concentration. The Stanbio

**Funding:** The author(s) received no specific funding for this work.

assay used in our laboratory is based on the glucose oxidase methodology adapted by Trinder et al. [2]. In this method, glucose is oxidized in the presence of glucose oxidase. After hydrogen peroxide is formed and reacts with phenol and 4-aminoantipyrine, a red-violet quinone complex forms. The intensity of the red-violet color is directly proportional to the glucose concentration (Stanbio Laboratory, Boerne, TX, USA). Though the laboratory method is the standard, it is more expensive and time consuming compared with the hand-held glucometer.

An alternative method is a hand-held electronic glucose measuring system designed for use in humans. This system uses electrochemical test strips in which blood is applied after the test strip is inserted into the glucometer. The blood is then drawn up the test strip via capillary action. Once it is in the glucometer, it reacts with glucose oxidase and forms gluconic acid. The gluconic acid then reacts with the test strip electrodes and creates an electrical current that is proportional to the concentration of glucose in the blood. This hand-held system has been validated for use in measuring glucose and β-hydroxybutyrate in dairy cows [3–6], however, this has not been validated for measuring glucose chute-side when performing a GTT in beef cattle. It is possible that beef cattle may have different response curves to a GTT (i.e. peak plasma glucose concentration, baseline plasma glucose concentration) and thus plasma glucose concentrations compared with dairy cattle, as it has been previously reported that plasma glucose and insulin concentrations were different between beef and dairy cows at the same stage of lactation [7].

The objective of the present study was to compare the glucose concentration using whole blood and the hand-held glucometer compared with the standard laboratory assay using plasma from samples obtained while completing a GTT on 13 Simmental × Angus steers during the growing phase. We hypothesized that the two different methods would show acceptable agreeance and that the hand-held meter would be a suitable method for quantifying glucose concentration chute-side.

## Materials and methods

All procedures were approved by The Ohio State University Institutional Animal Care and Use Committee (Animal Use Protocol # 2019A00000142).

### Animals and procedures

Thirteen Simmental × Angus steer calves were used for this glucose quantification comparison. Steers were trained in the chute 5 days/week for two weeks to allow for the steers to become acclimated to standing in the chute and being touched during the GTT.

Steers were fasted for 24 hours before the GTT. The morning of the GTT, steers were weighed to determine bolus size (0.25 g of glucose/kg BW delivered in a 50% weight/volume dextrose solution. After being weighed, jugular catheters were placed in the steers, and then steers were returned to their pen and allowed a one hour rest period before the GTT began. Blood samples were collected at 5 and 2 minutes before administration of the glucose bolus to determine fasted plasma glucose concentration. Subsequent blood samples were collected immediately after glucose bolus infusion (0 minutes), 5, 10, 15, 20, 30, 60, and 120 minutes after glucose bolus infusion. Before and after each 10 mL blood sample was collected, the catheter line was flushed with 4–5 mL of sterile heparinized saline (9 g/L of NaCl). All blood samples were transferred to a tube containing sodium fluoride and then immediately placed on ice. The sodium fluoride tubes were transferred back to the laboratory and centrifuged for 25 min at 2500 x g and 4°C. The plasma was then further aliquoted into individual microcentrifuge tubes to determine plasma glucose concentrations at a later date.

## Glucometer method

As blood was being sampled via the jugular catheter for the GTT, blood was immediately placed on the glucometer test strip to determine whole blood glucose concentration. The glucometer was used according to the label descriptions and directions of the manufacturer and is reported to measure blood glucose concentrations from 20–500 mg/dL (Precision Xtra; Abbott Diabetes Care Inc., Mississauga, ON, Canada). According to the user's manual, if a sample reads "LO", the meter has determined that the blood glucose concentration is below 20 mg/dL, and a sample reads "HI", the meter has determined that the blood glucose concentration is above 500 mg/dL. One sample that was measured with the glucometer returned a "HI" result, and the glucose concentration was recorded as 500 mg/dL as recommended by the glucometer's user manual. All samples measured with the glucometer were only measured once.

## Laboratory trinder method

The laboratory analysis was completed using a colorimetric assay (Stanbio Glucose LiquiColor (Oxidase) Procedure, Stanbio Laboratory, Boerne, TX, USA). Any sample that was outside of the linear portion of the standard curve was diluted with a 1:2 dilution. All samples were run in duplicates and the intra- and inter-assay coefficient of variations were 2.93% and 3.00%, respectively.

## Statistical analysis

First, using the raw measurements from the glucometer and the average of the duplicate laboratory samples, a paired t-test was completed using the TTEST procedure of SAS (SAS 9.4). Additionally, a Pearson's Correlation was completed for the two methods using the CORR procedure of SAS. For both of these models, data was assessed for normality using the residuals panel in SAS. When plotted, the differences between the pairs of observations were approximately normally distributed. Next, the Bland-Altman [8] approach was used to plot the difference between the two measurements against their mean to determine the two approaches' agreement. Bland and Altman [8] and Petrie and Watson [9] recommend first performing a paired t-test to test the null hypothesis that the mean of the differences between the two methods is zero, and that the differences are evenly scattered above and below zero. The paired t-test determined if there was evidence of a systematic difference between the hand-held glucometer and the laboratory assay. The next step is to perform a Pearson correlation, however, Bland and Altman [8] caution that a Pearson correlation coefficient only gives indication of how close the observations in the scatter diagram are to a straight line and do not assess agreement. To assess agreement, one needs to know how close the points are to the line of perfect agreement (the 45˚ line through the origin). The Bland-Altman plot determines the limits (± 1.96 standard deviations) within which 95% of the differences are expected to lie when the difference between the two measurements in a pair are plotted against their mean. If there is no evidence of a systematic effect, the points should be scattered evenly above and below the line corresponding to a zero difference. If the variability of the differences is not constant (i.e. funnel shape), Petrie and Watson [9] recommend transforming the data and repeating the process. If there is no evidence of a systematic effect in the Bland-Altman plot of either the raw or transformed data, the next step is to complete an index of agreement which can either be the intraclass correlation coefficient (ICC) or the Lin's concordance correlation coefficient (LCCC). These two indexes are similar and describe the closeness of the points to the line of perfect agreement and can be used to assess agreement because both accuracy and precision are incorporated. A Lin's concordance correlation coefficient was calculated to assess agreement between the two methods.

## Results

The paired t-test indicated no evidence of a systematic difference between the two methods of measurement with a test statistic of -0.86, ($P$ = 0.40). A Pearson correlation coefficient for the two methods of measurement was then completed on the raw data (Fig 1; $P$ < 0.001; r = 0.95), and indicated precision (i.e. the random variation describing the tightness of the points about the best-fitting straight line) between the two methods.

A Bland-Altman plot was completed for the raw data between the two measurements (Fig 2), with the mean concentration measured by the glucometer plotted against the difference in mean concentration between the two methods. The raw data demonstrated a funnel shape, indicating that the variability of the differences was not constant as the mean of the two measurements increased. Therefore, the raw data was log transformed as recommended by Petrie and Watson [9]. The transformed Bland-Altman plot is shown in Fig 3. Transforming the data eliminated the funnel effect, and the points are evenly scattered above and below the line representing the mean, corresponding to no systematic difference between the two methods. Additionally, since the scatter of the points is random with no funnel effect, we can conclude that the size of the discrepancy between the two methods is not related to the magnitude of the count. In Figs 2 and 3, the dashed lines represent the upper and lower limits of agreement. We expect 95% of the absolute differences to be less than the upper and lower limits of agreement, which is the case for the transformed data.

In addition, since there is no evidence of a systematic effect, we can estimate the Lin's concordance correlation coefficient. For this correlation coefficient, a perfect agreement is achieved when the value is equal to 1, and there is no agreement when it is equal to 0. For the

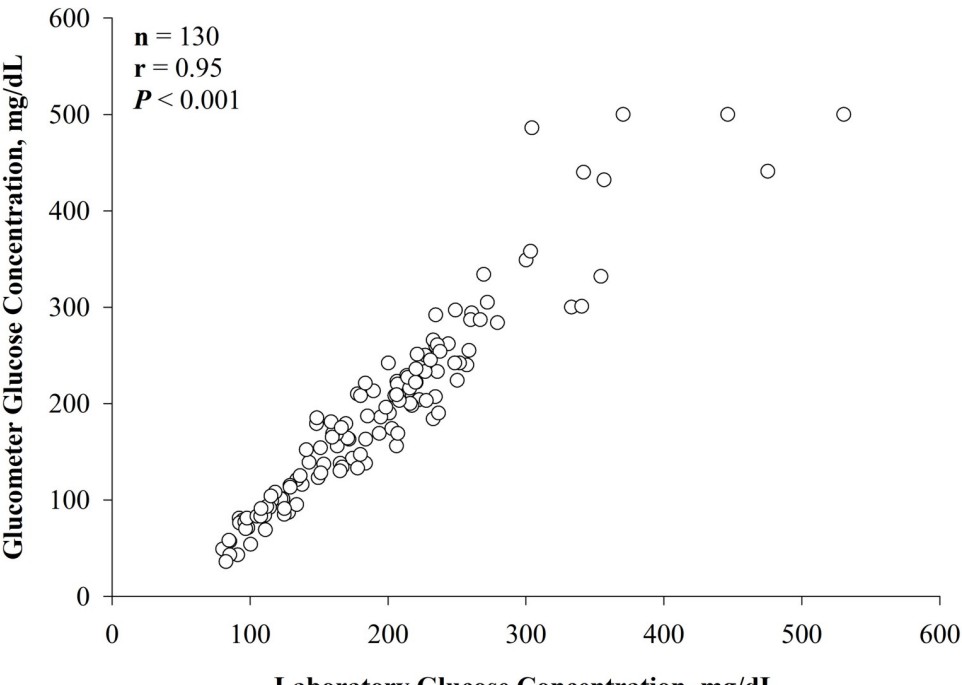

**Fig 1. Pearson correlation coefficient ($P$ < 0.001, r = 0.95) of the raw data (N = 130 samples from 13 steers), with the glucose concentration as measured by the glucometer on the y-axis (Precision Xtra; Abbott Diabetes Care, Inc., Mississauga, ON, Canada) plotted against the plasma preserved in sodium fluoride tubes and analyzed in a laboratory with a colorimetric assay (Stanbio Glucose LiquiColor (Oxidase) Procedure, Stanbio Laboratory, Boerne, TX, USA) on the x-axis.**

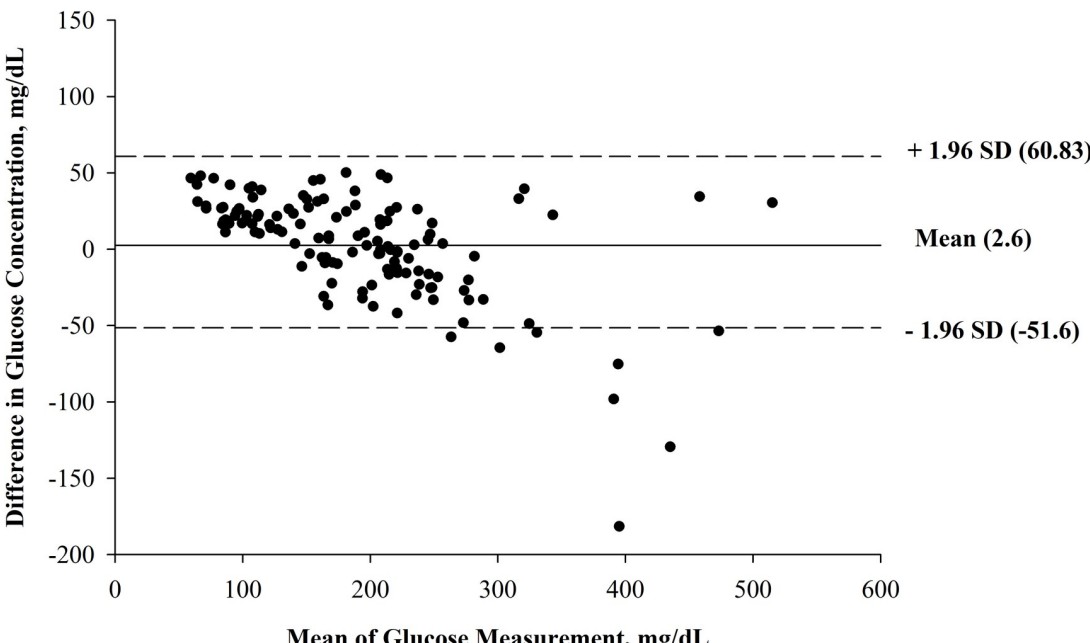

**Fig 2. Bland-Altman plot of the raw data (N = 130 samples from 13 steers), with the mean glucose concentration as measured by the glucometer (Precision Xtra; Abbott Diabetes Care, Inc., Mississauga, ON, Canada) and the plasma preserved in sodium fluoride tubes and analyzed in a laboratory with a colorimetric assay (Stanbio Glucose LiquiColor (Oxidase) Procedure, Stanbio Laboratory, Boerne, TX, USA) on the x-axis, plotted against the difference in glucose concentrations determined by the glucometer and the laboratory on the y-axis.** The mean difference is represented by the solid line (mean = 4.6) and the 95% confidence limits are represented by the dashed lines.

transformed data, the Lin's concordance correlation coefficient is 0.90 (95% confidence interval = 0.87–092). Petrie and Watson (2013) note that based on a previous article published by McBride et al. [10], a Lin's concordance correlation coefficient of $0.90 \leq r_c \leq 0.95$ is considered moderate agreement.

## Discussion

Taking into consideration the paired t-test, Pearson correlation coefficient, Bland-Altman plot, and Lin's concordance correlation coefficient, we accept our hypothesis that the handheld Precision Xtra glucometer moderately agrees with the laboratory method and is acceptable to use for rapid, chute-side measurement of glucose in beef cattle. Several studies have previously evaluated the agreement of the Precision Xtra glucometer with laboratory analysis of glucose in dairy cattle, however, this is the first study to our knowledge that has compared the two methods in beef cattle during a GTT [3–6].

Our results agree with those of previous reports that indicate the Precision Xtra glucometer has acceptable agreement with laboratory measurement in dairy cattle [3,5]. Both of these papers demonstrated Bland-Altman plots and reported that at least 95% of the observations fell within the 95% confidence intervals, indicating good agreement between the two methods. Neither paper, however, reported a Lin's concordance correlation coefficient or an intraclass correlation coefficient. Therefore, while there is agreement between the two methods based on the Bland-Altman plots, the assessment of agreement cannot be definitive without one of the indexes being calculated.

Additionally, of the papers that have previously reported unacceptable agreement between the Precision Xtra and laboratory analysis of glucose concentration only Lopes et al. [6]

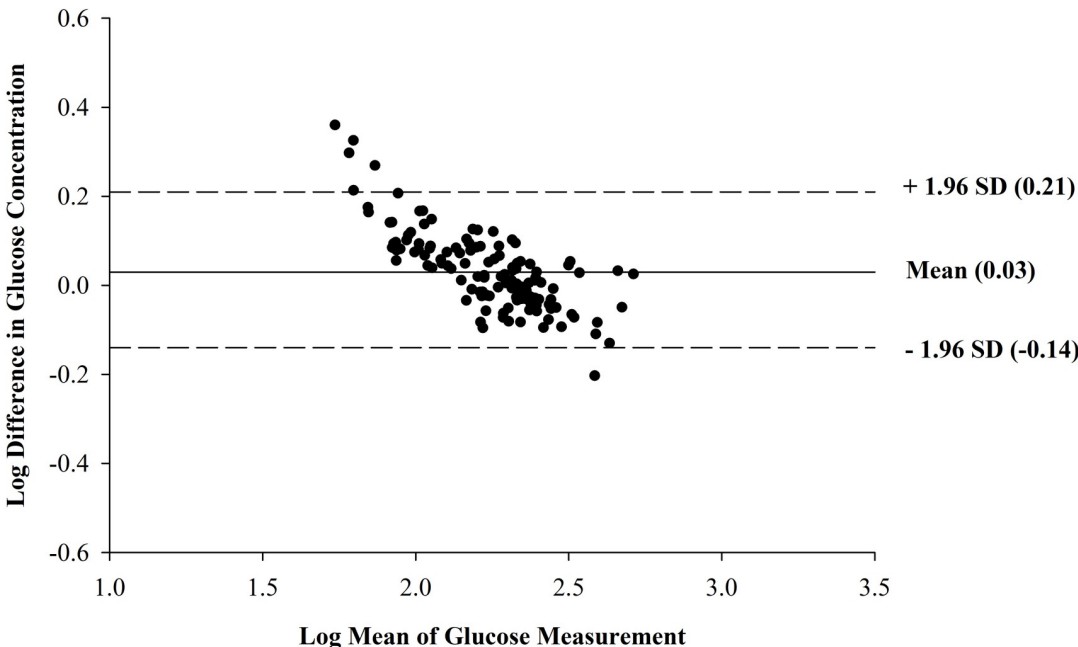

**Fig 3.** Bland-Altman plot of the log transformed data (N = 130 samples from 13 steers), with the mean glucose concentration as measured by the glucometer (Precision Xtra; Abbott Diabetes Care, Inc., Mississauga, ON, Canada) and the plasma preserved in sodium fluoride tubes and analyzed in a laboratory with a colorimetric assay (Stanbio Glucose LiquiColor (Oxidase) Procedure, Stanbio Laboratory, Boerne, TX, USA) on the x-axis, plotted against the difference in glucose concentrations determined by the glucometer and the laboratory on the y-axis. The mean difference is represented by the solid line (mean = 0.03) and the 95% confidence limits are represented by the dashed lines.

reported an LCCC or ICC while Megahed et al. [4] did not. It is interesting that Lopes et al. [6] showed moderate association between the two methods with a Pearson correlation coefficient (r = 0.71), indicated that their reported LCCC of 0.74 demonstrated strong agreement with the reference method, and reported that the difference between the methods fell within acceptable limits of agreement (± 1.96 standard deviations) at least 95% of the time, however, continued to state that according to the American Society for Veterinary Clinical Pathology guidelines, only 54.6% of the Precision Xtra glucose readings had a total observed error of ≤ 20% and declared the meter inadequate to measure glucose in dairy cows. In the present study, we did not measure total observed error, as Bland and Altman [8] and Petrie and Watson [9] do not discuss calculating this total observed error. Additionally, Petrie and Watson [9] follow the guidelines proposed by McBride [10] that state that a LCCC less than 0.90 is indicative of poor agreement between two methods. Following these guidelines, the LCCC of 0.74 reported to be in support of strong agreement should be observed with caution.

Similar to Zakian et al. [5], we used the glucose oxidase method for the laboratory glucose measurement. Using this method as the gold standard, we obtained similar results to Zakian et al. [5] and found the glucometer to be in moderate agreement to the laboratory measurement. As mentioned by Zakian et al. [5], one possibility for this improvement in agreement between the two methods may be the reference method used, as the other papers that evaluated the Precision Xtra glucometer compared its measurements to a hexokinase reference method [4,6]. Wittrock et al. [3] reported acceptable agreement between the two methods, however, did not mention which reference method was used and only stated that glucose concentrations were determined using a commercial reagent kit.

Of the previous studies that evaluated the agreement of the Precision Xtra glucometer with the laboratory measurements, only Wittrock et al. [3] similarly performed a GTT. Performing this method to assess agreement during a GTT is of great interest, as there are many samples that must be taken during this procedure and the hand-held glucometer could make performing the procedure quicker and cheaper if glucose concentration could be analyzed chute side. The present dextrose infusion that was provided to the beef steers (0.25 g of glucose/kg BW delivered in a 50% weight/volume dextrose solution) was the same as that used by Wittrock et al. [3] in dairy cows, however, the sampling timeline was different. Steers were sampled for blood in the current study at 5 and 2 minutes before glucose bolus infusion, and then subsequent samples were taken immediately after glucose bolus infusion (0 minutes) and then 5, 10, 15, 20, 30, 60, and 120 minutes after infusion. Wittrock et al. [3] sampled for blood immediately before dextrose infusion and then at 10 and 80 minutes after infusion. Wittrock et al. [3] reported that measurements that were within the physiological range of 2.3 to 5.2 mmol/L were slightly lower with the glucometer compared with the laboratory value. Alternatively, the authors found that the high glucose concentrations were generally overestimated by the glucometer, though the authors were not aware of a methodological reason for this difference. In the present study, we found that the glucometer and laboratory measurements were very precise and accurate up to concentrations $\leq$ 300 mg/dL. At glucose concentrations greater than 300 mg/dL, we observed our greatest differences between the glucometer and the laboratory measurement. Except for one blood sample that was obtained 5 minutes after glucose infusion, these high concentrations of glucose ($>$ 300 mg/dL) all coincided with our time 0 blood sample which was sampled immediately after glucose infusion. Generally, the glucometer overestimated glucose concentration when compared with the laboratory measurement at these high concentrations. This result is similar to that reported by Wittrock et al. [3]. However, when the concentrations greater than 300 mg/dL were removed from the analysis, the LCCC was only improved to 0.92 which still indicates moderate method agreement according to Petrie and Watson [9]. Therefore, we have only presented the statistical analyses including all of the data points from the GTT.

Since the full range of values included in the data set provided acceptable Bland-Altman plots and moderate agreement according to the LCCC, we conclude that the hand-held glucometer is acceptable to use for rapid, chute-side testing of blood glucose concentration in beef cattle. This glucometer was tested during a GTT, and while all of the data was included in the analyses presented, we caution its use at supraphysiological glucose concentrations such as that occurring immediately after glucose infusion during a GTT. However, it seems that under normal physiologic conditions the hand-held glucometer agrees with the laboratory glucose oxidase reference method. Additionally, based on our suppliers and current costs to complete a GTT for 13 steers, the glucometer method was 57% cheaper on a per sample basis compared with the laboratory method.

## Supporting information

**S1 Table.**
(XLSX)

## Acknowledgments

The authors would like to thank the staff at the Eastern Agricultural Research Station for all of their help in completing this project.

## Author Contributions

**Conceptualization:** Kirsten R. Nickles, Alvaro Garcia-Guerra, Francis L. Fluharty, Anthony J. Parker.

**Data curation:** Kirsten R. Nickles, Alejandro E. Relling, Anthony J. Parker.

**Formal analysis:** Kirsten R. Nickles, Alejandro E. Relling, Alvaro Garcia-Guerra, Anthony J. Parker.

**Investigation:** Kirsten R. Nickles, Alejandro E. Relling, Alvaro Garcia-Guerra, Francis L. Fluharty.

**Methodology:** Kirsten R. Nickles, Alejandro E. Relling, Francis L. Fluharty, Anthony J. Parker.

**Project administration:** Anthony J. Parker.

**Resources:** Anthony J. Parker.

**Supervision:** Alvaro Garcia-Guerra, Francis L. Fluharty, Anthony J. Parker.

**Validation:** Alvaro Garcia-Guerra.

**Visualization:** Kirsten R. Nickles, Anthony J. Parker.

**Writing – original draft:** Kirsten R. Nickles, Anthony J. Parker.

**Writing – review & editing:** Alejandro E. Relling, Alvaro Garcia-Guerra, Francis L. Fluharty, Anthony J. Parker.

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
