## [Decision Letter · Decision Letter 0]

23 Mar 2022

PONE-D-21-38258A comparison between two
glucose measurement methods in beef steers during a glucose tolerance
testPLOS ONE

Dear Dr. Parker,

Thank you for submitting your manuscript to PLOS ONE. After careful consideration, we
feel that it has merit but does not fully meet PLOS ONE’s publication criteria as it
currently stands. Therefore, we invite you to submit a revised version of the
manuscript that addresses the points raised during the review process.

Because one of the reviewers raised several crucial questions regarding the
hypothesis tested, methods and statistical analysis, I invite you to revise the
manuscript taken into consideration his comments. Also, please make sure to reply
all his comments with detailed answers to allow myself and the reviewer to better
understand what was proposed in the manuscript. Because we did not have many
comments from one of the reviewers at this time, and was quite difficult to find
available reviewers, I am passing the manuscript for you to review with the comments
made to avoid any further delay. However, after your revision the manuscript shall
be sent to a third reviewer for assessment. 

Please submit your revised manuscript by May 07 2022 11:59PM. If you will need more
time than this to complete your revisions, please reply to this message or contact
the journal office at plosone@plos.org. When
you're ready to submit your revision, log on to https://www.editorialmanager.com/pone/ and select the 'Submissions
Needing Revision' folder to locate your manuscript file.

Please include the following items when submitting your revised
manuscript:A rebuttal letter that responds to each point raised by the academic
editor and reviewer(s). You should upload this letter as a separate file
labeled 'Response to Reviewers'.A marked-up copy of your manuscript that highlights changes made to the
original version. You should upload this as a separate file labeled
'Revised Manuscript with Track Changes'.An unmarked version of your revised paper without tracked changes. You
should upload this as a separate file labeled 'Manuscript'.

If you would like to make changes to your financial disclosure, please include your
updated statement in your cover letter. Guidelines for resubmitting your figure
files are available below the reviewer comments at the end of this letter.

We look forward to receiving your revised manuscript.

Kind regards,

Marcio Duarte, PhD

Academic Editor

PLOS ONE

Journal Requirements:

Reviewers' comments:

Reviewer's Responses to Questions

**Comments to the Author**

1. Is the manuscript technically sound, and do the data support the conclusions?

Reviewer #1: Yes

Reviewer #2: No

2. Has the statistical analysis been performed
appropriately and rigorously? 

Reviewer #1: Yes

Reviewer #2: Yes

3. Have the authors made all data underlying the
findings in their manuscript fully available?

Reviewer #1: Yes

Reviewer #2: Yes

4. Is the manuscript presented in an intelligible
fashion and written in standard English?

Reviewer #1: Yes

Reviewer #2: Yes

5. Review Comments to the Author

Reviewer #1: This manuscript aimed to evaluate the effectiveness of a hand-held
glucometer in beef cattle The manuscript is within the scope of the Plos One and can
be accepted for publication. However, this manuscript should be classified as a
short communication.

Reviewer #2: Lines 56-59 - I really didn’t understand why you are pointing out these
limitations and possible errors of the laboratory method as a justification for
using the hand-held glucometer. Because you are using this method as a gold-standard
method for your comparison and recommendation of the alternative method.

Lines 60-62 - This sentence sounds very informal to a paper’s introduction. I believe
that should be in a more appropriate place, such as discussion.

Lines 68-70 - What is the real necessity to validate the same method for beef cattle?
Why the results should be different for beef cattle when compared with dairy cattle?
We are talking about the method.

Line 101 and line 104 – Is the range of reading from 20 to 300 or 20 to 500? There is
confusion here.

Lines 105-107 - This is not correct. This sample should be deleted from the database
because you are not certain about the value.

Lines 118-120 - Why did you not test for normality?? You need to be certain about
distribution because that could be biasing the results.

Lines 120-123 – In a general way, the information concerning statistical analysis and
methods for assessing adequacy were poorly described in this section. Some
explanations about these methods are dispersed over the results and discussion,
which is not the best place for that. I suggest that authors should be providing
these details in the current section.

Line 132 - Why did you not include regression to test the adequacy of your new
method?

Lines 137-139 - I am really not sure about the confidence to test transformed
variables. It is proved that variables transformation to follow Gaussian
distribution, do not represent a good estimate of the real value when a
back-transformation is applied. I believe that is a similar case. The funnel shape
is also present in the transformed data, but with lower intensity. The authors
should think about that.

Lines 142-1444 - It is clear that there is a bias over 300 mg/dL (Figures 1 and 2)
and I believe that should be analyzed and reported in the manuscript.

Lines 160-174 - This entire paragraph is closer to a material and methods than a
discussion.

Lines 215-218 - Which is the necessity to repeat this text?

6. PLOS authors have the option to publish the peer
review history of their article (what does this mean?). If published, this will
include your full peer review and any attached files.

If you choose “no”, your identity will remain anonymous but your review may still be
made public.

**Do you want your identity to be public for this peer review?** For
information about this choice, including consent withdrawal, please see our
Privacy Policy.

Reviewer #1: **Yes: **Otavio R. Machado Neto

Reviewer #2: No

---

## [Author Response · Author response to Decision Letter 0]

3 May 2022

I have attached a document titled response to reviewers comments that details the
author's response to both reviewers

to Reviewers PLOSone Glucose steers.docx
---

## [Decision Letter · Decision Letter 1]

6 Jul 2022

Short communication: A comparison between two glucose measurement methods in beef
steers during a glucose tolerance test

PONE-D-21-38258R1

Dear Dr. Parker,

We’re pleased to inform you that your manuscript has been judged scientifically
suitable for publication and will be formally accepted for publication once it meets
all outstanding technical requirements.

Kind regards,

Marcio Duarte, PhD

Academic Editor

PLOS ONE

Additional Editor Comments (optional):

Reviewers' comments:

Reviewer's Responses to Questions

**Comments to the Author**

1. If the authors have adequately addressed your comments raised in a previous round
of review and you feel that this manuscript is now acceptable for publication, you
may indicate that here to bypass the “Comments to the Author” section, enter your
conflict of interest statement in the “Confidential to Editor” section, and submit
your "Accept" recommendation.

Reviewer #2: All comments have been addressed

2. Is the manuscript technically sound, and do the data
support the conclusions?

Reviewer #2: Yes

3. Has the statistical analysis been performed
appropriately and rigorously? 

Reviewer #2: Yes

4. Have the authors made all data underlying the
findings in their manuscript fully available?

Reviewer #2: Yes

5. Is the manuscript presented in an intelligible
fashion and written in standard English?

Reviewer #2: Yes

6. Review Comments to the Author

Reviewer #2: The authors have answered all of my concerns. Because of that, I believe
the manuscript is now ready for publication.

7. PLOS authors have the option to publish the peer
review history of their article (what does this mean?). If published, this will
include your full peer review and any attached files.

If you choose “no”, your identity will remain anonymous but your review may still be
made public.

**Do you want your identity to be public for this peer review?** For
information about this choice, including consent withdrawal, please see our
Privacy Policy.

Reviewer #2: **Yes: **Alex Lopes da Silva

---

## [Editor Report · Acceptance letter]

11 Jul 2022

PONE-D-21-38258R1 

Short communication: A comparison between two glucose measurement methods in beef
steers during a glucose tolerance test 

Dear Dr. Parker:

I'm pleased to inform you that your manuscript has been deemed suitable for
publication in PLOS ONE. Congratulations! Your manuscript is now with our production
department. 

Kind regards, 

on behalf of

Dr. Marcio Duarte 

Academic Editor

PLOS ONE